# Fabrication and Characterization of Zein/Hydroxyapatite Composite Coatings for Biomedical Applications

**Yusra Ahmed, Muhammad Yasir and Muhammad Atiq Ur Rehman *** 

Department of Materials Science and Engineering, Institute of Space Technology Islamabad, 1, Islamabad Highway, Islamabad 44000, Pakistan; yusra0982@gmail.com (Y.A.); muhammadyasir85@gmail.com (M.Y.)
* Correspondence: muhammad.rehman.ur@fau.de

**Abstract:** Stainless steel is renowned for its wide use as a biomaterial, but its relatively high corrosion rate in physiological environments restricts many of its clinical applications. To overcome the corrosion resistance of stainless steel bio-implants in physiological environments and to improve its osseointegration behavior, we have developed a unique zein/hydroxyapatite (HA) composite coating on a stainless steel substrate by Electrophoretic Deposition (EPD). The EPD parameters were optimized using the Taguchi Design of experiments (DoE) approach. The EPD parameters, such as the concentration of bio-ceramic particles in the polymer solution, applied voltage and deposition time were optimized on stainless steel substrates by applying a mixed design orthogonal Taguchi array. The coatings were characterized by using scanning electron microscopy (SEM), energy-dispersive X-ray spectroscopy (EDX) and wettability studies. SEM images and EDX results indicated that the zein/HA coating was successfully deposited onto the stainless steel substrates. The wettability and roughness studies elucidated the mildly hydrophilic nature of the zein/HA coatings, which confirmed the suitability of the developed coatings for biomedical applications. Zein/HA coatings improved the corrosion resistance of bare 316L stainless steel. Moreover, zein/HA coatings showed strong adhesion with the 316L SS substrate for biomedical applications. Zein/HA developed dense HA crystals upon immersion in simulated body fluid, which confirmed the bone binding ability of the coatings. Thus the zein/HA coatings presented in this study have a strong potential to be considered for orthopedic applications.

**Keywords:** zein; hydroxyapatite; electrophoretic deposition; taguchi design of experiments

## 1. Introduction

According to the report published by the Global Opportunity Analysis and Industry Forecast (2017–2023), the global bio implant market value is anticipated to reach $124,000 million by the year 2023. This is primarily the result of the increase in chronic diseases and aging factors [1]. This increase in the market has resultantly boosted research in the area. Metallic bio-implants have been widely used in this market, as they offer various properties like strength, stability and biocompatibility. However, their corrosive nature in the physiological environment cannot be denied [2]. The resultant metallic ions from degradation in the body can prove to be fatal for body organs adjacent to implants [3]. After much research, the solution of surface treatment was proffered. Coating the implant with a biocompatible yet stable material can significantly enhance the corrosion resistance of the implant in the body. It further offers bone/implant integration, safety and efficacy [2]. Many biomaterials can be used as coating materials that significantly enhance bioactivity and biocompatibility [4,5]. A polymer such as zein has great attraction, owing to its unique properties. Zein is obtained from the endosperm of maize, which is composed of prolamins. It is a biocompatible polymer with applications in tissue engineering and drug delivery [6]. Owing to its low mechanical strength, it is mostly used with

inorganic composites such as bioactive glass (BG) 45S5 and hydroxyapatite (HA) [7]. HA is a widely used bioceramic for enhanced biocompatibility, with a chemical composition similar to that of natural bone [8,9]. There are several reliable techniques that offer HA coating, including radio frequency (RF) magnetron sputtering [10–12], plasma spraying [13–15], high velocity oxy-fuel plasma (HVOF) coating [16,17], the sol–gel process [18] and electrophoretic deposition (EPD) [19,20]. The optimum process is mainly selected on the basis of the coating material, substrate material and the coating thickness. It is said with surety that the final coating is strongly dependent on the fabrication process and the parameters opted for [21]. The literature elucidates that high temperature processes such as HVOF and Plasma spraying can alter the properties of metallic and ceramic substrates. The resultant change in composition and crystallinity of an HA coating can cause a decreased life span in a metallic implant [3]. Moreover, using thermal techniques to coat a low melting point polymer such as zein cannot be considered [22]. Similarly, RF Sputtering has been widely used for bio ceramic coatings such as HA [10], Si-CaP [11], Hopeite [23] and TiN [24], etc. The main limitation is the requirement of post-deposition annealing due to the formation of coatings with low crystallinity character [25]. In this article, the Electrophoretic deposition (EPD) technique is used to deposit the zein/HA coatings. It is a type of colloidal deposition based on the principles of electrophoresis, which deposits charged particles onto substrates, requires little equipment, and is cost-effective and facile [26]. EPD's applications encompass a variety of functional biomaterial coatings, thin films, scaffolds, drug delivery systems and biosensors, etc. Extensive studies have been performed on the topic to prove that this technique is best suited for HA coatings due to good stoichiometric control, high control on the purity of the deposited coating and the possibility of deposition on complex-shaped substrates, which is not easy to achieve from any other technique [27]. Moreover, it offers feasibility at low temperatures, and is hence best suited for the deposition of polymers with low melting points [28]. Usually, researchers use the 'hit and trial' method to optimize EPD parameters, which in most cases results in extra time consumption and material wastage [29]. For this reason, the design of experiments approach (DoE) was utilized, which allows a small number of experiments to reach the desired parameters [30]. In the EPD process, the DoE approach has been used in numerous studies e.g., to optimize BG/PEEK/Al$_2$O$_3$ [31], Chitosan/BG [32,33], Chitosan/HA [20] and Chitosan/gelatin/BG [29] on a variety of substrates. Thus, the Taguchi approach is being increasingly used in industries and labs for economical optimization [34]. Using this, the robustness of the process is calculated as a signal-to-noise (S/N) ratio. The assessment of this ratio is performed in optimization, which results in reproducibility and integrity [20]. Zein/BG coatings have been studied in the literature [35], which resulted in favorable osteogenic properties. Mariotti et al.'s study on zein-fiber mats incorporated with BG and Cu-doped BG particles resulted in improved osteogenic and angiogenic properties [36]. However, the stability of BG in the physiological medium is one of the issues which needs to be addressed. Therefore, zein/HA coatings have been put forward in the current study. Zein/HA coatings can significantly improve biocompatibility and stability in a physiological environment. Moreover, zein/HA coatings, being more stable, are expected to provide improved corrosion and scratch resistance. Zhi-Hu Qu et al. studied the combination of zein/HA scaffolds for bone tissue engineering, which proved to be an optimal biomaterial for said application [37]. Similar results were presented by Lian et al. [38] and Babaei et al. [39]. Here, we present zein/HA coatings deposited on 316L SS for the first time.

In this study, we obtain zein/HA coatings on 316L stainless steel (SS) via EPD. The EPD and suspension related parameters were optimized by Taguchi DoE. Scanning electron microscopy (SEM) images confirmed that fairly homogenous coatings were obtained from the optimum set of parameters. Energy-dispersive X-ray spectroscopy (EDX) analysis confirmed the presence of zein and HA in the coatings. The wettability studies elucidated the mild hydrophilic nature of the zein/HA coatings, which confirmed the suitability of the developed coatings for biomedical applications. Zein/HA coatings developed HA crystals on the surface of the coatings. Zein/HA coatings showed good adhesion with the substrate. To the best of the authors' knowledge, zein/HA coatings deposited via EPD have not

been investigated in the literature. Moreover, it is expected that the combination of zein and HA will improve the biocompatibility of the coatings.

## 2. Materials and Method

### 2.1. Materials

Zein powder (CAS number 9010-66-6) was obtained from SolarBio® (Beijing, China), and Hydroxyapatite (HA) powder with mean particle size 10 μm was used. Ethanol and acetic acid were obtained from Sigma Aldrich Chemie® (Schnelldorf, Germany). A 316L stainless steel foil (1 mm thick) was used for substrates.

### 2.2. Suspension

Zein powder (6 wt %) was added to ethanol (74 wt %) and distilled water (20 wt %) in a 100 mL bottle, and the solution was magnetically stirred using a hot plate for 30 min at 37 °C, followed by stirring for another 30 min at room temperature. The pH was maintained at a value of 3.5–4 by adding acetic acid dropwise (approx. 10 mL). Later, HA particles (in a concentration of 1.25 g/L and 5 g/L) were introduced, followed by magnetic stirring for 30 min. Finally, the suspension was ultrasonicated for one hour, which allowed the preparation of a stable suspension of zein/HA prior to the EPD process. The procedure for preparing a stable suspension of zein based composite was adopted from the studies performed by Rivera et al. [35,40] and Kaya and Boccacini [41].

### 2.3. EPD Coating

AISI 316L stainless steel electrodes (dimensions of 30 × 25 mm) were cleaned with a mixture of acetone and ethanol, and then rinsed with deionized water and dried. The electrodes were kept parallel at a distance of 10 mm. A direct current (DC) cathodic EPD process was performed at different voltages and deposition times, as shown in Table 2. The zeta potential of suspended particles is one of the critical parameters of EPD, and has important effects on both the deposition rate and the stability of the suspension. Zein molecules acquire a positive charge when dispersed in the aqueous ethanol solution at a pH of ~5.3 due to the protonation of carboxyl and amine groups [35,41]. Similarly, HA dissolves best in aqueous ethanol at pH 3–5 [19]. According to experiments performed by Baştan et al., HA particles have a positive charge at a pH lower than 6 [19]. This explains the cathodic deposition of the zein/HA particles, as observed in this study (pH was maintained at 4–4.5).

### 2.4. Optimization of EPD Parameters and Designing the Array

To form the Taguchi array of experiments for the optimization of EPD parameters, a mixed Taguchi array was constructed from 3 control factors (concentration of HA in the suspension, voltage and time), out of which concentration had 2 levels, whereas voltage and time had 4 levels each, as illustrated in Table 1. The software (Minitab 18 ™, Beijing, China) gave an array of 8 experimental runs, as shown in Table 2. Each experiment was repeated thrice. The samples were weighed using a digital balance before and after the coating, and average values were assessed.

**Table 1.** Selected factors and levels.

| Symbol | Control Factor | Levels 1 | 2 | 3 | 4 |
|--------|----------------|----------|---|---|---|
| **A** | Concentration of HA in the suspension (g/L) | 1.25 | 5 | | |
| **B** | Voltage (V) | 3 | 5 | 7 | 9 |
| **C** | Time (min) | 6 | 9 | 12 | 15 |

**Table 2.** Mixed design Array.

| Concentration | Time | Voltage |
|:---:|:---:|:---:|
| 1.25 | 3 | 6 |
| 1.25 | 5 | 9 |
| 1.25 | 7 | 12 |
| 1.25 | 9 | 15 |
| 5 | 3 | 15 |
| 5 | 5 | 12 |
| 5 | 7 | 9 |
| 5 | 9 | 6 |

According to the literature, if the concentration of HA particles in the suspension is increased to 7.5 g/L, the particles sediment, i.e., the suspension becomes unstable. In addition to this, high voltages lead to the generation of gas bubbles during the EPD process [20]. According to the literature [40], the zein coatings obtained at lower voltages (3–5 V) are very thin and transparent. Therefore, voltages lower than 3 V were not selected. A good deposition of zein was shown at 10 V.

*2.5. Characterization of the Coatings*

The microstructure at the surface was examined by scanning electron microscope (SEM, Mira3 Tescan, TESCAN, Kohoutovice, Czech Republic) at an energy of 10 kV. Energy-dispersive X-ray spectroscopy (EDX) was performed at 15 kV (LEO 435VP, Carl Zeiss™ AG, Oberkochen, Germany).

The stability of the zein/HA coatings was assessed by the zeta potential measurements (Malvern Instruments, Malvern, UK).

Corrosion studies were carried out by immersing the coatings in Dulbecco's modified eagle medium (DMEM) at 37 °C and plotting the dynamic polarization curve at a scan rate of 3 mV s$^{-1}$, in the potential range from $E_{corr}$ − 500 mV to $E_{corr}$ + 500 mV (IM6eX Xpot potentiostat, Zahner Elektrik GmbH, Kronach, Germany). A three-electrode system was used, with Pt as a counter electrode, Ag/AgCl as a reference electrode and the substrate to be tested as a working electrode. The contact angle between the surface of the coatings and a droplet of deionized water (volume of droplet = 5 μL) was measured. In order to carry out the test, a fixed volume of deionized water was dropped on the surface of the coatings, and digital images were taken after every five-seconds until the total time lap was 30 s. The values of the contact angle obtained after every five seconds were averaged out, and the mean value was reported along with the standard deviation. The test was repeated five times for each of the characterized samples. The surface roughness of the zein/HA coatings was tracked by means of a laser profilometer (UBM, ISC2). In order to conduct the test, a straight line of 5 mm was drawn on each sample. Then, the machine was given a command to measure the average roughness across the drawn line at a scan speed of 400 points per second. The inbuilt software, UBM, was used to calculate the average surface roughness of the samples. It is important to mention that the biocompatibility of the implant is often determined by the surface roughness, wettability and surface chemistry. It is a combination of these three factors which determines the suitability of the implanted biomedical device. Thus, these factors determined the performance of the biomedical surface in a physiological environment.

The adhesion strength of the zein/HA composite coatings deposited via EPD on 316L SS was evaluated by scratch tests. In order to perform a scratch test, the sample was prepared with a surface area of 1 cm². The scratch tester was tuned in to the Revetest mode (CSM™ Revetest machine, Needham, MA, USA). The scratch test was carried out, keeping in view the maximum adhesion strength of the coatings, i.e., load/force was increased incrementally from 1 N to 20 N. In order to apply the load, a diamond indenter with a diameter of ~200 μm was applied on the Rockwell-hardness testing

machine. The diamond indenter indented a continuous scratch in the form a line. The length of the line was kept at 5 mm, and the indenter speed was 5.2 mm/min. Three different tracks were performed on each sample. Critical loads were determined through acoustic emission (AE) measurements and optical observations of the scratch. Two critical loads ($L_{C1}$ and $L_{C2}$) were determined for each specimen: the first critical load ($L_{C1}$) corresponds to the load at which the first crack occurred, while the second ($L_{C2}$) corresponds to the load at which the coating was completely removed from the scratch channel.

The ability of the implant to bond with the host tissues, i.e., the bone, is often evaluated by placing the samples in a solution with an ionic composition close to that of human blood plasma. One of the important physiological solutions with an ionic concentration closer to that of human blood plasm is simulated body fluid (SBF). The composition and method of preparation have been reported in the literature [41]. The coated samples were sectioned into surface areas of 225 mm². The coated samples were then placed in bottles already filled with 50 mL of SBF. The coated samples were then put into an incubator for 3 days. The incubated sample was taken out and analyzed via SEM to identify the changes in the morphology of the coatings. This procedure is termed 'in-vitro bioactivity evaluation', which is considered the qualitative representation for the bone binding feature of the implants.

## 3. Results and Discussion

### 3.1. Suspension Stability

The details about the effect of deposition kinetics and mechanisms on the morphology and thickness of zein based composite coatings have been discussed in the literature [4]. To deposit zein/HA coatings, stable suspensions with a zeta potential of +25 ± 5 mV, +30 ± 5 mV at a pH of 4–4.5 were prepared, containing 5 g/L HA and 1.25 g/L HA in the zein solution (prepared earlier), respectively. The relatively high value of zeta potential indicates the stability of the suspension, which is important to obtain uniform coatings by EPD. The EPD mechanism of zein based composites has been explained by [33,42]. Uncharged zein is insoluble in water and organic solvents. However, protonated zein dissolves in water-ethanol-acetic acid mixtures at a low pH (pH < 5). During the application of an electric field (EPD process), positively charged zein molecules move towards the cathode, lose their charge and form an insoluble deposit at the cathode. Moreover, HA is positively charged under acidic pH; hence, it is expected to move towards the cathode and deposit by coagulation. The electrophoretic mobility of HA particles is much higher than that of zein. Therefore, the low concentration of zein co-deposition of zein/HA is expected. However, higher concentrations of HA will lead to a substantial decrease in the electrophoretic mobility of zein (due to the increase in pH and conductivity). These findings formed the basis for choosing the concentration of HA (1.25–5 g/L) in this study.

### 3.2. Design of Experiments Study

This method allows the study of each parameter at various levels by averaging deposition yield, standard deviation and corresponding S/N ratios at each level. The deposition yield, S/N ratio for deposition yield and standard deviation in deposition yield were calculated using the following formulae [30]:

$$Deposition\ yield = \frac{\Delta\ Weight}{A}\ in\ \left(\frac{mg}{cm^2}\right) \tag{1}$$

where $\Delta\ Weight$ = weight after coating – weight before coating and $A$ = Area of coating.

$$\frac{S}{N} ratio\ of\ deposition\ yield = -10 \log\left[\frac{1}{n}\left(\sum 1/y^2\right)\right] \tag{2}$$

where $y$ = *deposition yield* and $n$ = number of observations.

$$\frac{S}{N} ratio\ of\ deposition\ yield = -10 \log\left[\frac{1}{n}\left(\sum y^2\right)\right] \tag{3}$$

where $y$ = standard deviation and $n$ = no. of observations.

For the mixed design array, deposition yield, standard deviation and respective S/N ratios were calculated, and the results are reported in Table 3.

**Table 3.** Experimentally calculated deposition yield and corresponding standard deviation.

| Control Factors | | | Deposition Yield | S/N Ratio | Standard Deviation | S/N Ratio |
|---|---|---|---|---|---|---|
| Concentration (g/L) | Time (min) | Voltage (V) | (mg/cm$^2$) | (dB) | | (dB) |
| 1.25 | 3 | 6 | 0.23333 | 1.716821 | 0.158605 | 20.80852 |
| 1.25 | 5 | 9 | 0.093333 | −6.24198 | 0.041096 | 32.53885 |
| 1.25 | 7 | 12 | 0.346667 | 5.155527 | 0.322215 | 14.65196 |
| 1.25 | 9 | 15 | 0.033333 | −15.1851 | 0.009428 | 45.32639 |
| 5 | 3 | 15 | 0.7 | 11.25925 | 0.058878 | 29.41574 |
| 5 | 5 | 12 | 0.44 | 7.226339 | 0.08641 | 26.08359 |
| 5 | 7 | 9 | 0.50667 | 8.451732 | 0.094281 | 25.32639 |
| 5 | 9 | 6 | 0.49333 | 8.220095 | 0.033993 | 34.18695 |

The response values of the mean of the mean deposition yield, the signal to noise ratio (S/N) of the deposition yield, the mean of standard deviation and the S/N of the standard deviation deposition were calculated, as given in Tables 4–7, respectively. The Δ (maximum–minimum) value for each control factor was also calculated. The highest value of Δ indicates that the given control factor is the most significant among all [29].

**Table 4.** Mean of mean response for the deposition yield of zein/HA coatings.

| Factors | | 1 | 2 | 3 | 4 | Δ (Maximum − Minimum) |
|---|---|---|---|---|---|---|
| **Conc. (g/L)** | **A** | 0.535 | 0.176667 | | | 0.358333 |
| **Time (min)** | **B** | 0.466667 | 0.266667 | 0.426667 | 0.263333 | 0.203333 |
| **Voltage (V)** | **C** | 0.366667 | 0.393333 | 0.3 | 0.363333 | 0.093333 |

**Table 5.** S/N response for the deposition yield of zein/HA coatings.

| Factors | | 1 | 2 | 3 | 4 | Δ (Maximum − Minimum) |
|---|---|---|---|---|---|---|
| **Conc. (g/L)** | **A** | 8.789353 | −3.63869 | | | 12.42805 |
| **Time (min)** | **B** | 6.488034 | 0.49218 | 6.80363 | −3.48252 | 6.31145 |
| **Voltage (V)** | **C** | −1.96295 | 6.190933 | 1.104877 | 4.968458 | 5.086057 |

**Table 6.** Mean of the deposition yield's standard deviation.

| Factors | | 1 | 2 | 3 | 4 | Δ (Maximum − Minimum) |
|---|---|---|---|---|---|---|
| **Conc. (g/L)** | **A** | 0.068391 | 0.132836 | | | 0.064445 |
| **Time (min)** | **B** | 0.108742 | 0.063753 | 0.208248 | 0.021711 | 0.186537 |
| **Voltage (V)** | **C** | 0.034153 | 0.204312 | 0.067688 | 0.096299 | 0.170159 |

**Table 7.** S/N response for the standard deviation.

| Factors | | 1 | 2 | 3 | 4 | Δ (Maximum − Minimum) |
|---|---|---|---|---|---|---|
| **Conc. (g/L)** | **A** | 28.75317 | 28.33143 | | | 0.421739 |
| **Time (min)** | **B** | 25.11213 | 20.36778 | 19.98917 | 39.75667 | 19.7675 |
| **Voltage (V)** | **C** | 37.37106 | 20.36778 | 28.93262 | 27.49774 | 17.00329 |

Tables 4 and 5 indicate that the deposition yield of the zein/HA coatings is more sensitive to the changes in concentration of HA in the suspension, as the Δ (maximum − minimum) value obtained from the control factor A, i.e., the concentration of HA in the suspension, is the maximum. The reason

could be that the concentration of the particles in the suspension changes the deposition kinetics and allows the deposition of more HA (as more HA was available in the suspension). Similar results were reported in a study of HA composite based coatings by A. Pawlik et al. [20]. Moreover, Tables 6 and 7 illustrated that the standard deviation in deposition yield is more sensitive to changes in the deposition time (similarly reported in [29]). Therefore, it can be concluded that concentration is the most significant factor, with high statistical confidence in the data.

The effect of control factors on deposition yield is plotted (Figure 1) in Tables 4–7.

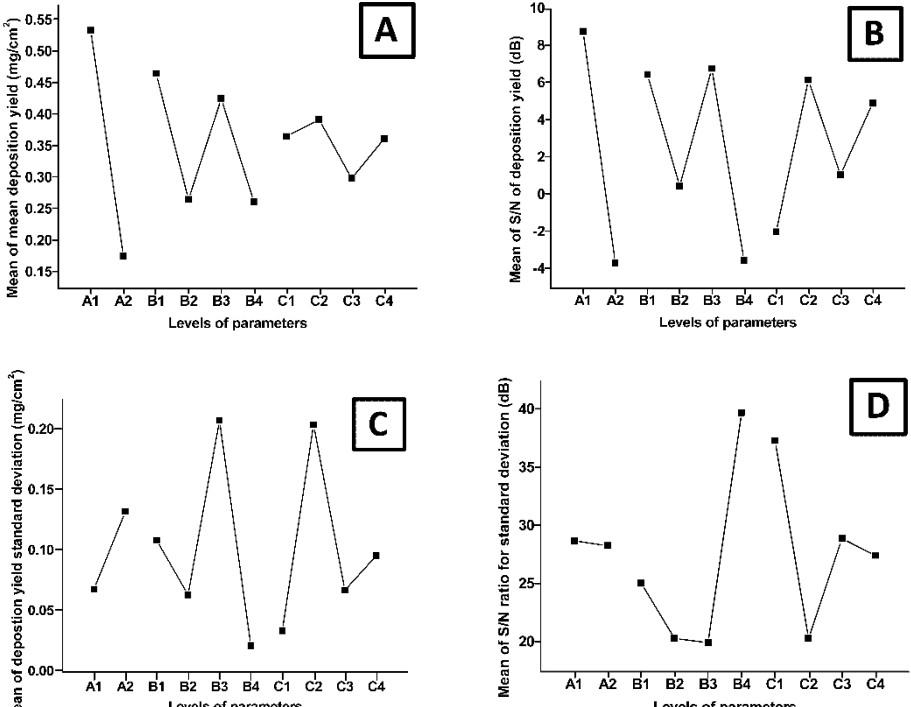

**Figure 1.** Effect of control factors on deposition yield. (**A**) Mean of mean deposition yield; (**B**) S/N ratio for deposition yield; (**C**) mean of standard deviation; (**D**) S/N ratio for standard deviation.

It was concluded from Figure 1A,B that that maximum (best) deposition yield would be obtained at parameters A1 (5 g/L of HA in the suspension), B1 (3 min) and C2 (12 V). The S/N response for deposition yield is highest for A1, B1 and C2. This shows that there no difference between the two responses. The increase in deposition yield with an increase in time and voltage for the 5 g/L concentration experiments is in accordance with Hamaker's law [30]. Figure 1C,D indicated that the best parameters, on the basis of the standard deviation, are A1 (5 g/L), B1 (3 min) and C1 (15 V). It can be concluded from Figure 1 that the best parameters for depositing zein/HA coatings via EPD are the A1, B1 and C2 parameters (based on the combination of the deposition yield and standard deviation), which showed the highest deposition yield with the lowest standard deviation [29].

To calculate the statistical significance of different factors on deposition yield (the dependent variable), a multivariate analysis of variance (MANOVA) was conducted, as depicted in Table 8. The results confirm that the concentration factor is the most significant among the three factors, with the least probability ($p \leq 0.05$); similar results of significance were drawn from Tables 4 and 5.

**Table 8.** MANOVA analysis effect of control factors on the deposition yield of zein/HA coatings made by EPD.

| Control Factor | Concentration | Time | Voltage |
|---|---|---|---|
| *p*-value | 0.050 | 0.421 | 0.814 |

### 3.3. Morphological Analysis

Figure 2A–C shows the SEM images of the zein/HA coatings (obtained via EPD, with electric field of 15 V applied for 3 min by keeping the concentration of HA at 5 g/L) at different magnifications. The characteristic structure of zein, i.e., a homogenous porous film, was observed. Similar results have been reported for the zein based coatings obtained via EPD [43]. The HA particles are integrated and dispersed fairly homogeneously into the pores of the zein matrix. The pores are partially blocked by the incorporation of agglomerated HA, which improves the mechanical stability of the film.

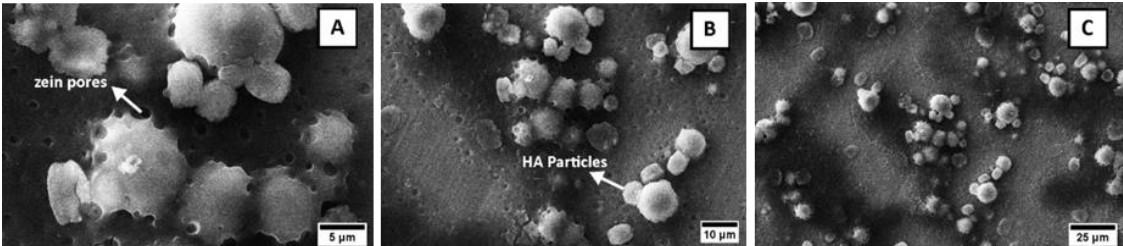

**Figure 2.** SEM images of a zein/HA coating on stainless steel, obtained via vertical standing EPD at different magnifications (parameters: 5 g/L of HA in the suspension, 3 min, 15 V); (**A**) at 5μm resolution, (**B**) at 10 μm resolution, (**C**) at 25 μm resolution.

### 3.4. Composition Analysis (Qualitative)

The area EDX analysis of Figure 3A confirms the presence of both materials; peaks relevant to the HA, i.e., Ca and P, were observed in the EDX spectra [20]. Furthermore, the presence of the N and C peaks may indicate the presence of zein [44]. However, the detection of C via EDX is not accurate [30].

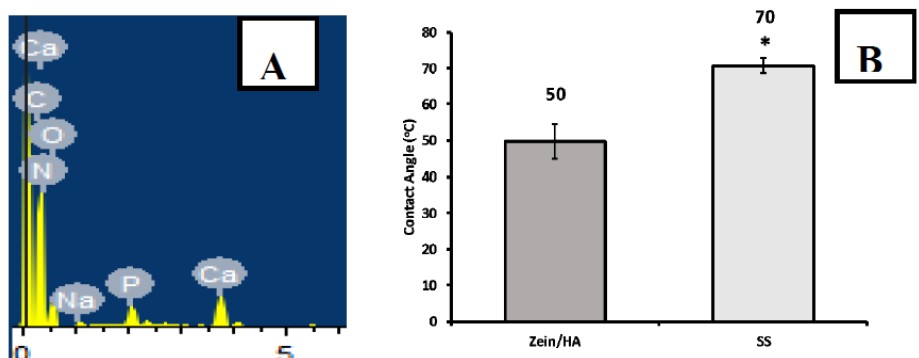

**Figure 3.** (**A**) EDX Spectrum; (**B**) Contact Angle comparison with a stainless steel substrate of zein/HA coatings obtained via EPD (parameters: 5 g/L HA in the suspension, 3 min, 15 V) (* significant at $p < 0.05$). The data represents the mean ± standard deviation of five experiments for each system.

### 3.5. Wettability Studies

The measured contact angle of zein/HA coatings (obtained at a deposition voltage of 15 V, with a deposition time of 3 min and 5 g/L HA in the suspension) was measured to be 50 ± 2°, which is near the ideal wettability value of 35–80° for the initial protein attachment [45]. Figure 3B shows a strong decrease in the value of the contact angle of the zein/HA coating in comparison to that of the stainless-steel substrate. A statistical *t*-test was conducted to determine the statistical difference between the wettability of the zein/HA coating and bare stainless steel ($p = 0.0301$). Thus, zein/HA coatings improved the surface properties of stainless steel. Zein is amphiphilic in character [44]. Its hydrophobicity is due to the presence of amino acids (such as proline, leucine, isoleucine and alanine), whereas its hydrophilicity is due to glutamine [46]. HA is hydrophilic owing to the presence of hydroxyl groups [47]. The HA incorporation enhances the hydrophilic nature of zein [46], and thus

the combination of both leads to the overall hydrophilic nature of the coating. Moreover, it may also be possible that the glutamine chains were present at the top of the coatings, which may have led to the hydrophilicity in the zein/HA coatings deposited via EPD.

### 3.6. Roughness Measurements

In addition to wettability, surface topography is also important in determining the response of the surface toward the cellular interaction. Various studies have suggested that surface roughness, wettability and surface chemistry should be critically analyzed in order to design a surface which is favorable for the attachment of proteins and the representative cells [48]. The surface topography was evaluated in terms of the average surface roughness of the zein/HA coatings. The average surface roughness ($R_a$) of zein/HA composite coatings was $1 \pm 0.1$ μm, and maximum roughness ($R_{max}$) was measured to be $3 \pm 0.2$ μm. The mean roughness of the zein/HA coatings was in the range suitable for the attachment and proliferation of osteoblast-like cells and bone-marrow-derived ST-2 cells [9,29,43]. The statistically significant difference was confirmed using a *t*-test ($p = 0.020$), as shown in Figure 4. Surface topography, wettability and surface chemistry are the basic building blocks for the enhanced interaction between the implant surface and the human body. The surface properties are more important in biological applications, because the true bulk material may never see the physiological environment. Therefore, it is the interface between the implant and the host tissue which will determine the fate of the implanted biomedical device in terms of interaction at the cellular and mechanic levels.

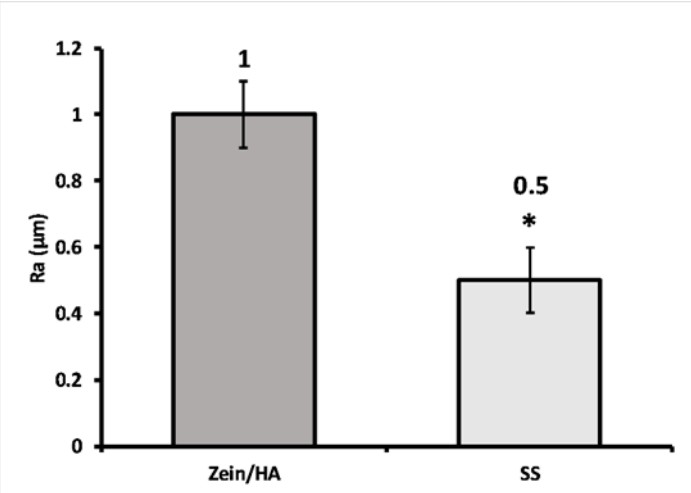

**Figure 4.** Ra value comparison of zein/HA coating and bare SS with a mean ± standard deviation for five samples of each system (* marks the statistically significant difference $p < 0.05$).

### 3.7. Corrosion Studies

The ability of the zein/HA coatings to inhibit the release of metallic ions from the metallic substrate was tracked using a potentiodynamic polarization scan, as shown in Figure 5. It was inferred from Figure 5 that the stainless steel samples tended to passivate with an increase in the applied potential. This result means that the corrosion process will slow down due to the formation of a chromium oxide layer, which is highly stable and prevents further corrosion from occuring. In contrast, zein/HA coatings did not passivate upon increasing the applied potential. From the view of the cathodic branch of the polarization curve, it was observed that the zein/HA significantly reduced the cathodic current densities. The corrosion current density was calculated by applying a Tafel fit to the polarization curve (Figure 5). The corrosion current density for the 316L SS sample was 0.32 μA/cm$^2$, and for the zein/HA coatings, the value was 0.046 μA/cm$^2$. The lowest corrosion current density was achieved by the zein coatings, i.e., 0.033 μA/cm$^2$. It was observed that the current density for zein/HA coatings

was slightly higher than that of the zein coatings. The reason for the slightly higher current density of the zein/HA compared to that of the zein could be that the HA particles may be at higher energy levels, thus raising their tendency to corrode. The difference in corrosion current density between zein and zein/HA coatings was statistically insignificant. The zein/HA coatings show a good corrosion protection effect, as the EPD process leads to the formation of a biostable film of zein reinforced with HA particles, which effectively acts as a barrier on the metallic substrate.

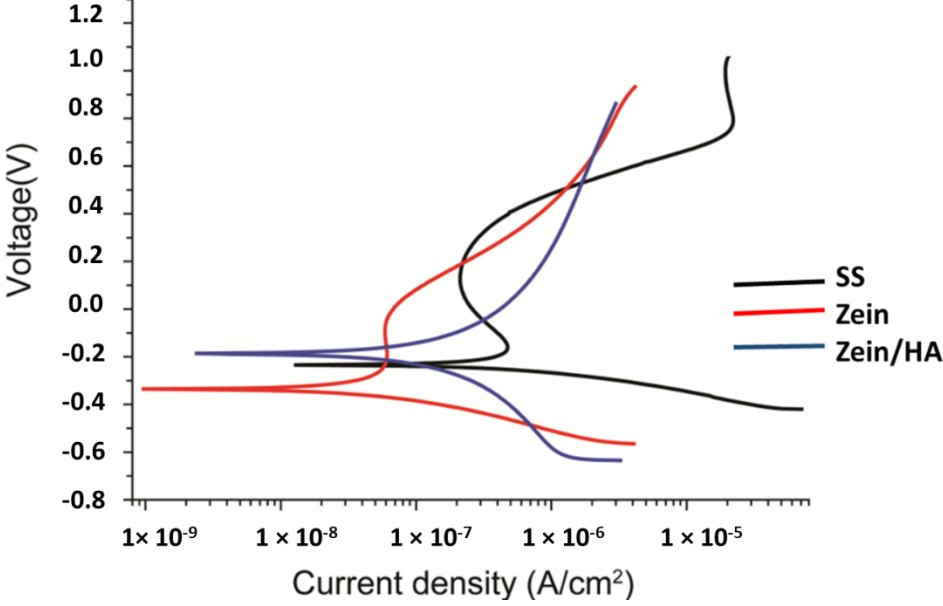

**Figure 5.** Polarization curves in DMEM at 37 °C for bare SS, zein and zein/HA coatings.

### 3.8. Adhesion Strength (Scratch Test)

The adhesion strength of zein/HA coatings with the stainless steel substrate was measured quantitatively by a scratch test. The results of the scratch test showed that the value of the lower critical load ($Lc_1$) for zein/HA coatings was ~4 N. Once the load was further increased and reached the value of 7 N, the zein/HA coatings presented circular cracks on the surface of the coatings. The measured values for the second critical load were ~10 N, which indicated that the zein/HA had developed good mechanical linkage with the surface of the substrate. The overall result confirmed that the adhesion strength between the zein/HA coatings and the substrate was sufficient for the design of orthopedic implants [19,29,49].

### 3.9. In-Vitro Bioactivity

SEM micrographs (Figure 6) show the change in the morphology of the coating surface after immersion in SBF, characterized by the formation of pores and a nanostructured apatite layer. Figure 6A shows that a cauliflower-like structure covered the surface of the zein/HA coating after 3 days of immersion in SBF. This cauliflower-like structure is an indication of the formation of HA crystals, which confirms the bioactive nature of the zein/HA coatings [4,6,11,43]. Figure 6B elucidates that the HA formed on the surface of the zein/HA coatings is plate-like [40], which indicates the formation of a calcium-enriched apatite layer [4], since the iso-electric point (IEP) of the zein/HA is at a lower pH range than the pH of SBF. This suggests that the zein/HA coatings will be negatively charged in the SBF, and there will be a strong possibility for attracting calcium ions from the SBF. Thus, the apatite layer formed on the surface of the zein/HA coatings was found to be enriched with a calcium casing, forming a plate-like morphology on the top surface of zein/HA coatings.

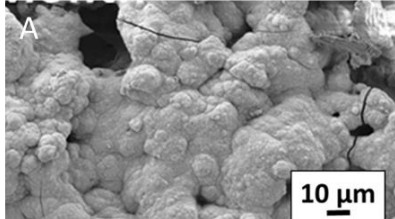 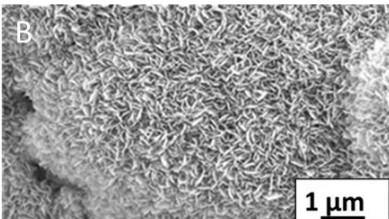

**Figure 6.** SEM images of zein/HA coatings after treatment in SBF for 3 days. (**A**) Low magnification and (**B**) higher magnification.

## 4. Conclusions

In this study, the EPD of zein/HA was performed on a stainless steel substrate using a Taguchi DoE approach. The parameters of the concentration of HA particles, time and voltage were optimized in relation to the deposition yield of the resultant coatings. It is concluded that the optimum parameters for depositing zein/HA coatings on 316L SS are: a concentration of HA in the suspension of 5 g/L, a deposition time of 3 min and an applied voltage of 12 V. One set of the experiments was performed at parameters of concentration: 5 g/L, time: 3 min and voltage: 15 V, which is closest to the optimal parameters. An SEM, an EDX and a wettability analysis were performed on the obtained zein/HA coatings on 316L SS substrates. SEM images showed that the porous structure of zein was partially incorporated with HA particles, which indicates the improved mechanical stability of the zein/HA coatings. The EDX result confirmed the presence of zein and HA in the obtained coatings. Moreover, the zein/HA coatings showed appropriate wettability (a contact angle of 50°) for the initial protein attachment. Zein/HA showed the appropriate roughness values for cell attachment and proliferation. Zein/HA coatings improve the corrosion resistance of bare 316L SS. Thus, zein/HA coatings offer a protective effect and can prevent the release of toxic metallic ions from 316L SS in a physiological environment. Moreover, zein/HA adhered strongly with the 316L SS substrate (an adhesion strength of ~10 N between the coatings and substrate). The coatings were bioactive upon immersion in SBF. It can be concluded that the zein/HA coatings developed in this study have the potential to be considered for further in vivo studies. The zein/HA composite coatings, developed for the first time via EPD, presented favorable mechanical, biological and surface properties for biomedical applications. Moreover, the qualitative adhesion strength of the zein-based biocompatible coating was elucidated for the first time.

**Author Contributions:** Conceptualization, M.A.U.R. and Y.A.; methodology, Y.A.; software, Y.A.; validation, Y.A., M.A.U.R. and M.Y.; formal analysis, Y.A. and M.Y.; investigation, Y.A., M.Y. and M.A.U.R.; resources, M.A.U.R.; data curation, Y.A.; writing—original draft preparation, Y.A.; writing—review and editing, M.Y. and M.A.U.R.; visualization, Y.A. and M.A.U.R.; supervision, M.Y. and M.A.U.R.; project administration, M.A.U.R. and Y.A.; funding acquisition, M.A.U.R. All authors have read and agreed to the published version of the manuscript.

**Funding:** This research received no external funding.

**Acknowledgments:** Sherbano Zaidi from Institute of Space Technology Islamabad, Pakistan for helping in the experiments and procurement, Fatih Erdem Bastan fromSakarya University Istanbul, Turkey for providing HA.

**Conflicts of Interest:** The authors declare no conflict of interests.

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
