# Peer review of "Fabrication and Characterization of Zein/Hydroxyapatite Composite Coatings for Biomedical Applications"

_surfaces, doi:10.3390/surfaces3020018_

Round 1

Reviewer 1 Report

The authors presented an interesting study on developing zein/hydroxyapatite composite coatings on stainless steel substrate by Electrophoretic Deposition. The paper is well structured by presenting successively the introduction along with the experimental method followed by results, discussion, and conclusion. However, there are some minor comments and corrections, addressing which can improve the work presentation:

  1. The motivation of the research is not well elaborated in the abstract. Please point out the gaps and problems that you are addressing before explaining your approach.
  2. The state of the art could be more updated and comprehensive. In the beginning, it is nice to briefly point out coating methods using the following references.
  • Review of geometries and coating materials in solid phase microextraction: opportunities, limitations, and future perspectives. (2017) Analytica chimica acta984, 42-65.
  • Coating techniques for functional enhancement of metal implants for bone replacement: A review. (2019) Materials12(11), 1795.
  1. Also, the following important articles are missed regarding zein HA coatings:
  • Lian, H., Liu, X., & Meng, Z. (2019). Enhanced mechanical and osteogenic differentiation performance of hydroxyapatite/zein composite for bone tissue engineering. Journal of Materials Science54(1), 719-729.
  • Babaei, M., Ghaee, A., & Nourmohammadi, J. (2019). Poly (sodium 4-styrene sulfonate)-modified hydroxyapatite nanoparticles in zein-based scaffold as a drug carrier for vancomycin. Materials Science and Engineering: C100, 874-885.

  1. Fix the cross-references in the text that are shown as “Error! Reference source not found.”
  2. Please present the ANOVA table along with p-values for parameters in your Taguchi analysis.
  3. In Taguchi S/N plots, usually, although different levels of on parameter are connected, different parameters are shown separated from each other. Please fix this in figure 1 for clarification.
  4. At the beginning of the conclusion, in 2, 3 sentences, restate the problem and briefly mention the approach you employed before stating the results.
  5. A proper improvement in English is required.

Author Response

We are thankful to reviewers for the detailed comments. The manuscript has been revised accordingly (please see the attached file). We hope that the manuscript is now acceptable for publication.

Reviewer 2 Report

  1. Better comparison with other methods is suggested. e.g. plasma-spraying , RF magnetron sputtering to prepare hydroxyapatite-based coatings are used. The authors may refer to the works of professors Surmenev R. in respect with RF-magnetron sputtering works.
  2. The reproducibility of the results presented should be described in more details. Statistical analysis should be done in all the cases to compare the results obtained and reveal the difference.
  3. Did the authors obtain some results of surface energy values? This allows to get more reliable and complete information on the properties of the surfaces obtained.

Author Response

(The authors gave the same response as above.)

Round 2

Reviewer 2 Report

Can be accepted